# Gird Based Line Segment Detector and Application: Vision System for Autonomous Ship Small Assembly Line

Jinhong Ding [1,2] and Chongben Ni [1,2,*]

1   State Key Laboratory of Ocean Engineering, Shanghai Jiao Tong University, Shanghai 200240, China;
    ahaha@sjtu.edu.cn
2   School of Naval Architecture, Ocean & Civil Engineering, Shanghai Jiao Tong University,
    Shanghai 200240, China
*   Correspondence: nichongben@sjtu.edu.cn

**Abstract:** The shipbuilding industry demands intelligent robot, which is capable of various tasks without laborious pre-teaching or programming. Vision system guided robots could be a solution for autonomous working. This paper introduces the principle and technique details of a vision system that guides welding robots in ship small assembly production. TOF sensors are employed to collect spatial points of workpieces. Huge data amount and complex topology bring great difficulty in the reconstruction of small assemblies. A new unsupervised line segment detector is proposed to reconstruct ship small assemblies from spatial points. Verified using data from actual manufacturing, the method of this paper demonstrated good robustness which is a great advantage for industrial applications. This paper's work has been implemented in shipyards and shows good commercial potential. Intelligent, flexible industrial robots could be implemented with the findings of this study, which will push forward intelligent manufacturing in the shipbuilding industry.

**Keywords:** intelligent manufacturing; machine vision; autonomous robotic welding; point cloud; line segment detector

## 1. Introduction

An industrial robot is the technical trend of intelligent shipbuilding and is expected to replace labor work in actual manufacturing. However, most welding jobs in shipyards are still performed manually because conventional welding robots lack flexibility. Industrial robots could be classified into 2 categories: teach-replay robots and offline-programming robots, which trajectories are achieved through laborious manual teaching or programming. These approaches are impractical in the shipbuilding industry because shipbuilding consists of multi-type and small-batch manufacturing for which the time and labor cost of manual teaching or programming are unacceptable from a commercial point of view. Thus most welding jobs in shipyards are still performed by manual operators. As rising labor cost and lack of skilled workers are posing challenges to shipyards, the demands of intelligent welding robot, which is capable of autonomous welding, is proposed by the shipbuilding industry.

Studies of autonomous robotic welding in shipbuilding have been attempted [1–5]. The major drawback in previous works is that robots' trajectories depend on guide rails or rollers, thus repetitive installation and uninstallation of rails or robots consumes too much labor and time. Programming of welding robots also leads unfavorable ration of programming time to production time. Approaches to reduce robot programming costs could be classified in 3 ways: CAD-based method, hybrid method, and vision-based method. The CAD-based method employs macros or templates to reduce programming efforts [6,7]. This leads to significant maintenance costs and does not consider the derivation of workpieces. The hybrid method combines the CAD model with visual images, enjoying better flexibility but still requires CAD data or manual input [8]. The vision-based method

could be an optimized solution for intelligent robotic welding because it is completely independent of CAD data and routine user inputs [9,10]. Meanwhile, the vision-based method proposes a great challenge, and splitting it into several small tasks could be a better solution [11]. Chen described robot pose control for the weld trajectory of a spatial corrugated web sheet based on laser sensing [12]. Additionally, welding robots with vision-based weld seam tracking modular are developed, which enjoy better tolerance to position deviation than conventional robots [13–18]. Research aiming at weld seam recognition are also conducted. Tsai produced welding path plans for golf club heads [19], and Zhang reconstructed a single weld seam using structured light [20]. Tsai and Zhang's work successfully recognize a single weld seam in laboratory conditions. However, in previous studies, the problem was not solved under actual manufacturing conditions because workpieces in shipyards usually include complicated weld seam structures. The aforementioned studies did not consider multiple workpieces with various shapes, which is common in actual manufacturing in the shipyard.

Grid-based algorithms are generally more computationally efficient than other algorithms [21]. Most grid-based algorithms achieve a time complexity of O(n), where n is the number of spatial points. STING [22], Wave Cluster [23], and CLIQUE [24] are the most commonly used grid-based algorithms. The aforementioned algorithms emphasize point clustering more than line segment detection. However, the topic of the present study was intelligent robotic welding for shipbuilding, which requires segments reconstructed from complicated spatial points.

For the demand of intelligent shipbuilding, a vision-based method is presented to achieve autonomous robotic welding for small assemblies of various shapes without pre-teaching or programming. This method is tested in actual manufacturing data from the shipyard and demonstrated good tolerance, robustness, and accuracy.

## 2. Hardware Implementation

This study focuses on autonomous welding of small assemblies. Small assembly refers to the basic components in shipbuilding, consisting of plates and stiffeners, as in Figure 1. Most small assemblies are limited within the dimensional size of 4 m × 4 m, and the weight of 1 ton. All stiffeners need to be double-sided welded onto the plates.

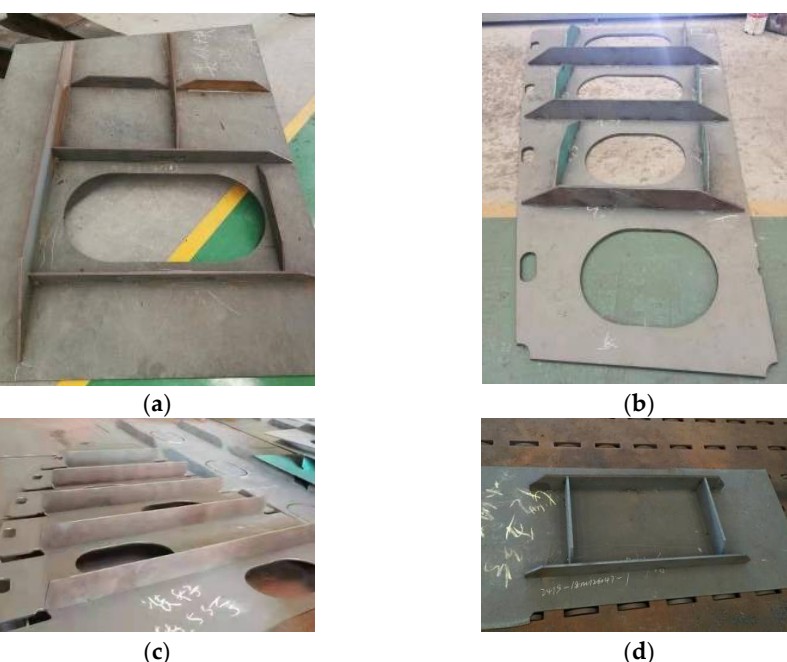

(a)

(b)

(c)

(d)

**Figure 1.** Examples of small assemblies in the shipyard. (**a**,**b**,**d**) consist of orthogonal stiffeners, and (**c**) consists of parallel stiffeners.

The hardware of this study is displayed in Figure 2. Welding robots and time-of-flight (TOF) laser sensors are installed on the gantry. The robot controller and TOF sensors are connected to the PC by LAN cable.

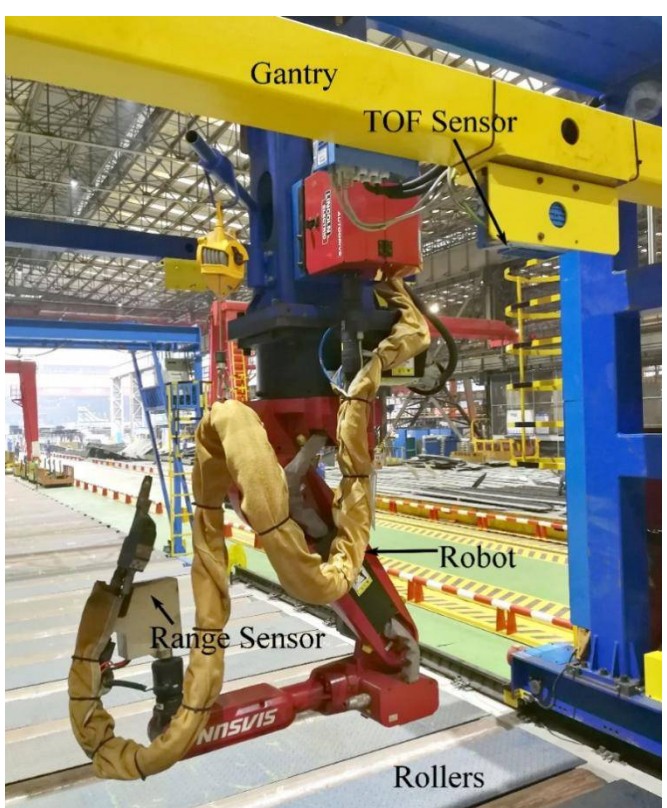

(**a**)

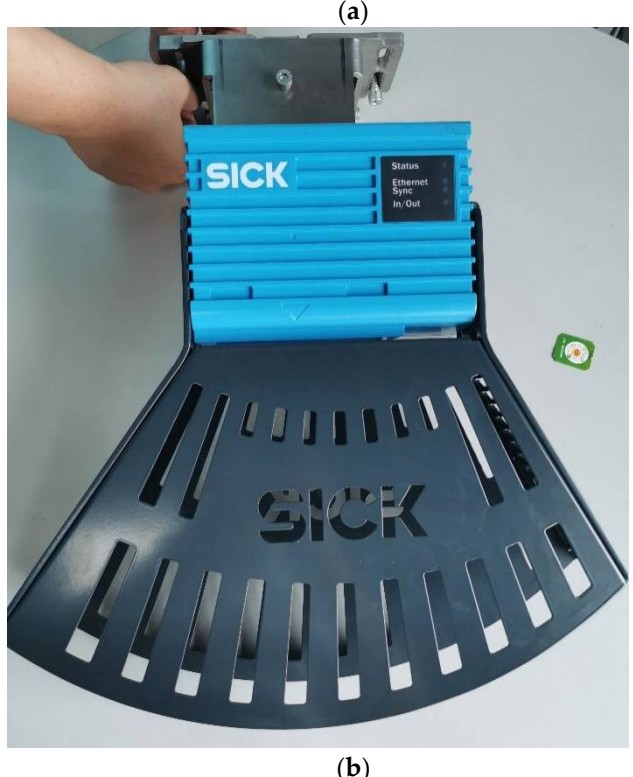

(**b**)

**Figure 2.** (**a**) Robot and TOF sensor (shroud removed) installed on a gantry; (**b**) TOF sensor (with shroud).

The TOF laser sensors cast laser stripes onto small assemblies and collect depth values. As the gantry moves, the TOF laser sensors scan over small assemblies on rollers, as shown in Figure 3.

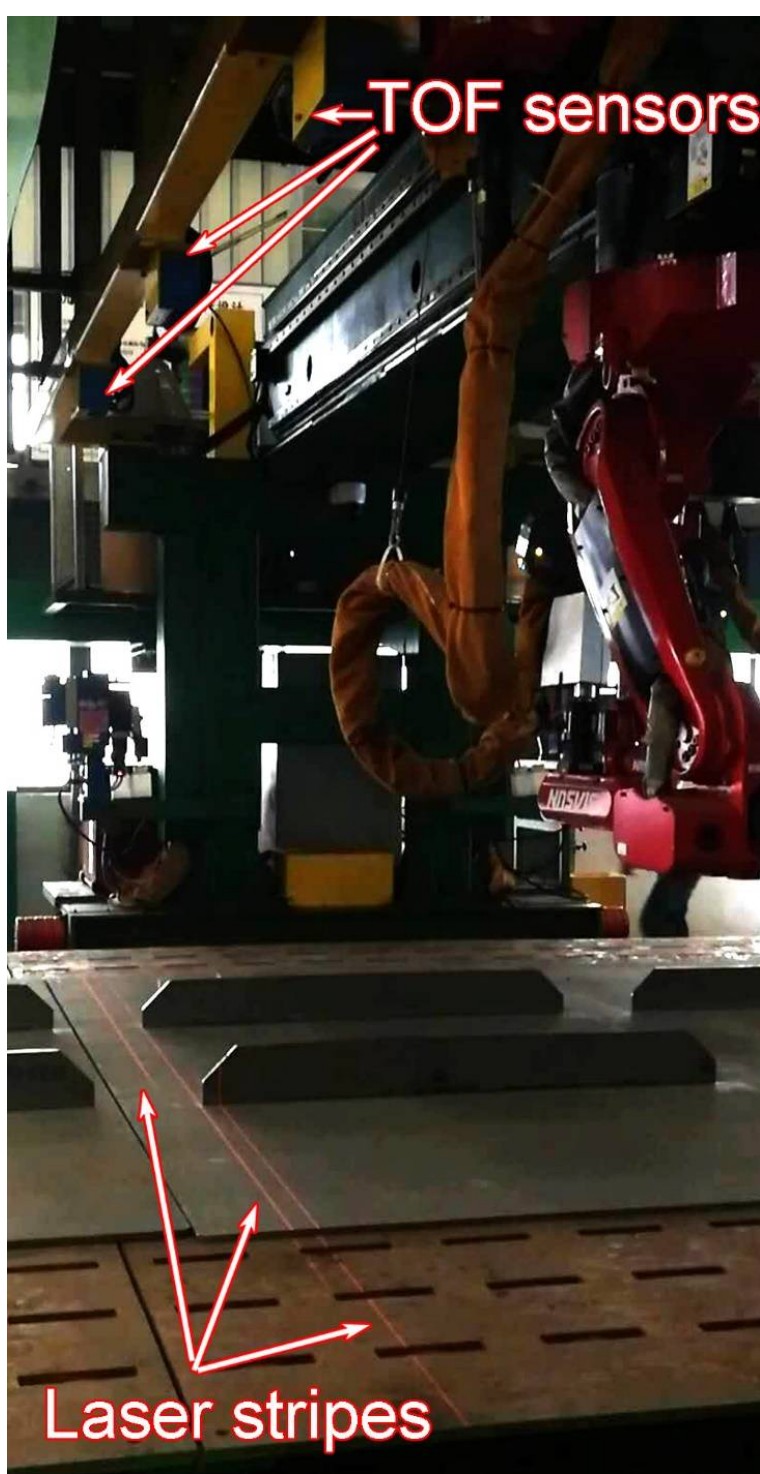

**Figure 3.** TOF sensors scan over small assemblies.

Data from the TOF laser sensor are organized in a depth matrix. The depth matrix is visualized in a depth-colored grayscale image, as shown in Figure 4. It could be transformed into a point cloud of the workpiece as the scan parameters are provided.

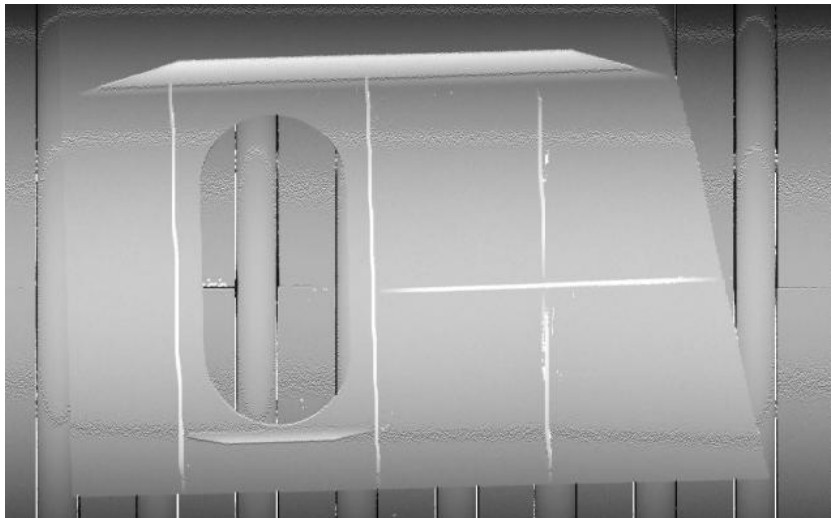

**Figure 4.** Depth-colored grayscale image of small assemblies.

### 3. Reconstruction of Small Assembly

Reconstruction of small assembly is the precondition for autonomous robotic welding. The welding robot needs the exact location of each weld seam, which could be calculated from the stiffener's centerline. Thus, the section introduces the method adopted in the reconstruction of all the stiffeners' center lines from the depth matrix.

#### 3.1. Spatial Points of Stiffeners

To reduce computation cost, the depth matrix is processed using various approaches to separate profiles from the background. For instance, Zhang used the Canny edge detector and Tsai calculated the geometric center of profiles [19,20]. In this study, we adopted Tsai's method because it preserves profile information of stiffener. In this method, the recognition of the profile is performed within each scanned point of each laser stripe. Figure 5a shows a TOF sensor casting a laser stripe onto a workpiece with 2 stiffeners, and Figure 5b displays scanned points collected by the TOF sensor, in which the convex bumps in the dashed circle clearly indicate the location of the stiffener profile.

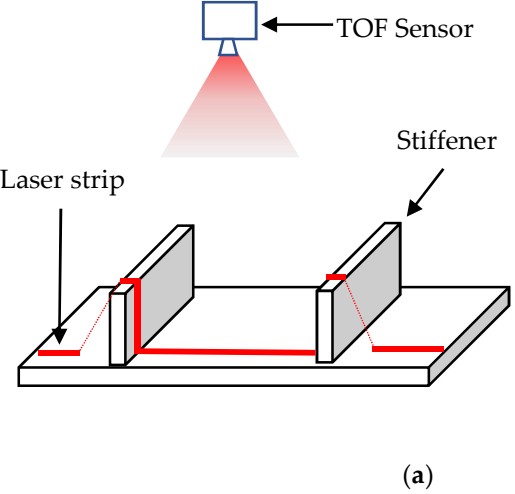

(**a**)

**Figure 5.** *Cont*.

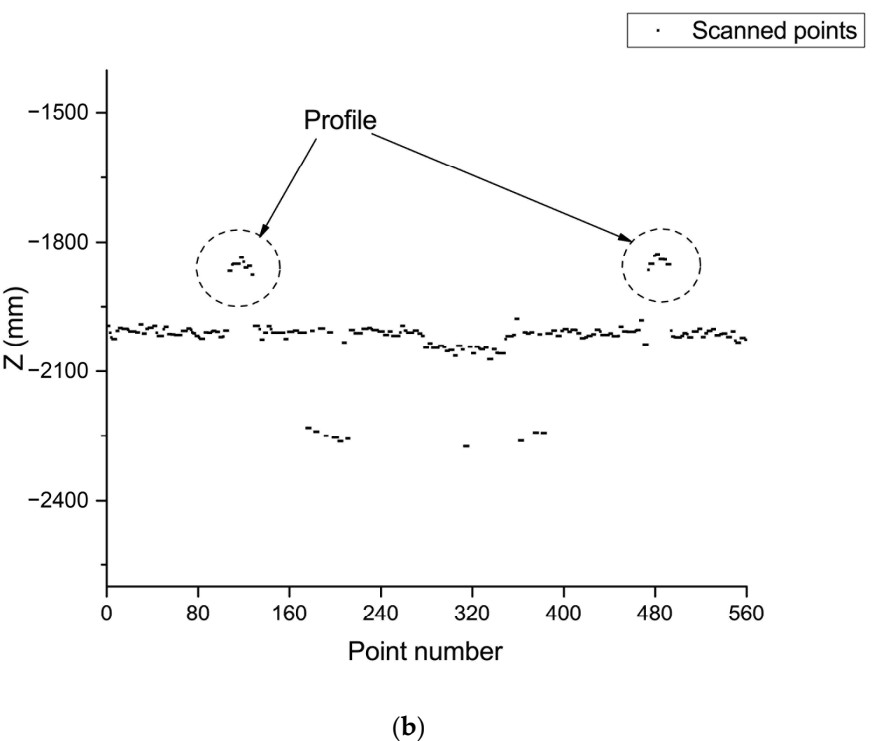

(**b**)

**Figure 5.** (**a**) Laser stripe on the small assembly; (**b**) Profiles in the row of scanned points.

By extracting the profile centers from each column, the spatial points of stiffeners are obtained. Figure 6 gives examples of small assemblies and corresponding spatial points in this study.

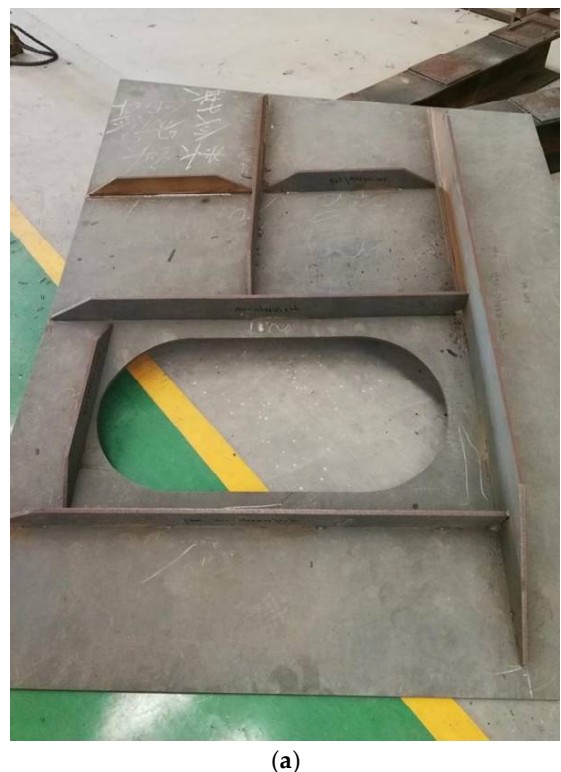

(**a**)

**Figure 6.** *Cont.*

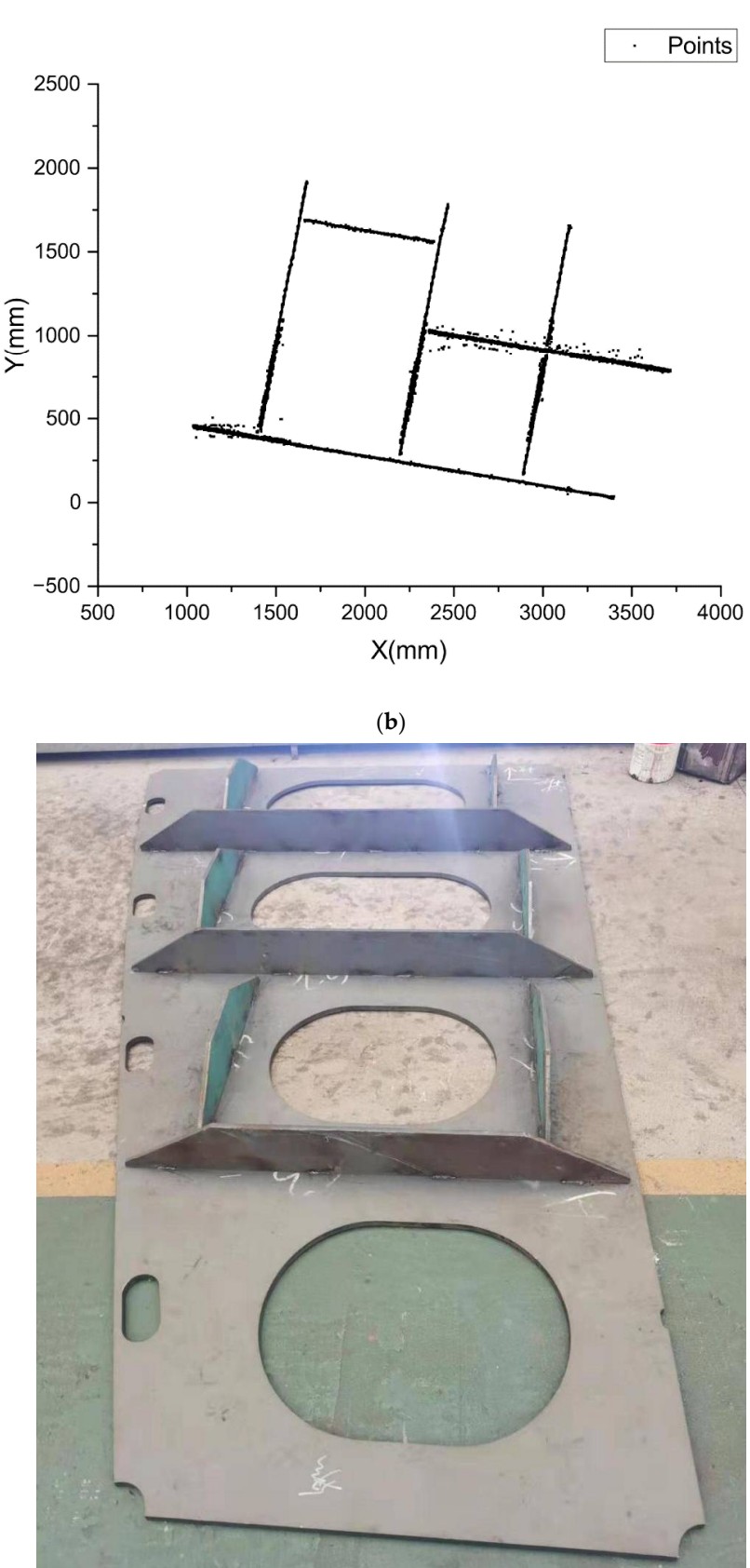

(**b**)

(**c**)

**Figure 6.** *Cont.*

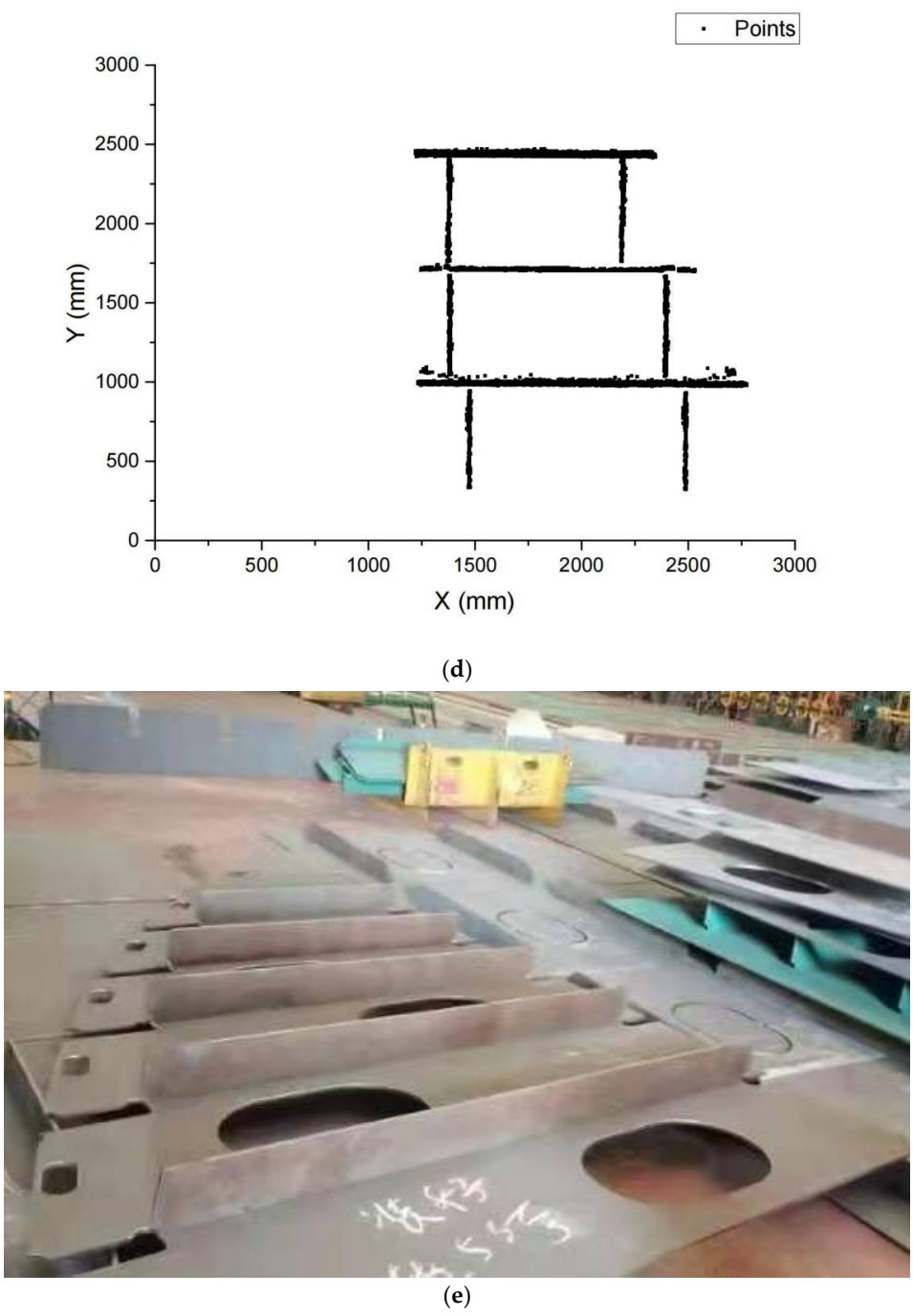

(**d**)

(**e**)

**Figure 6.** *Cont.*

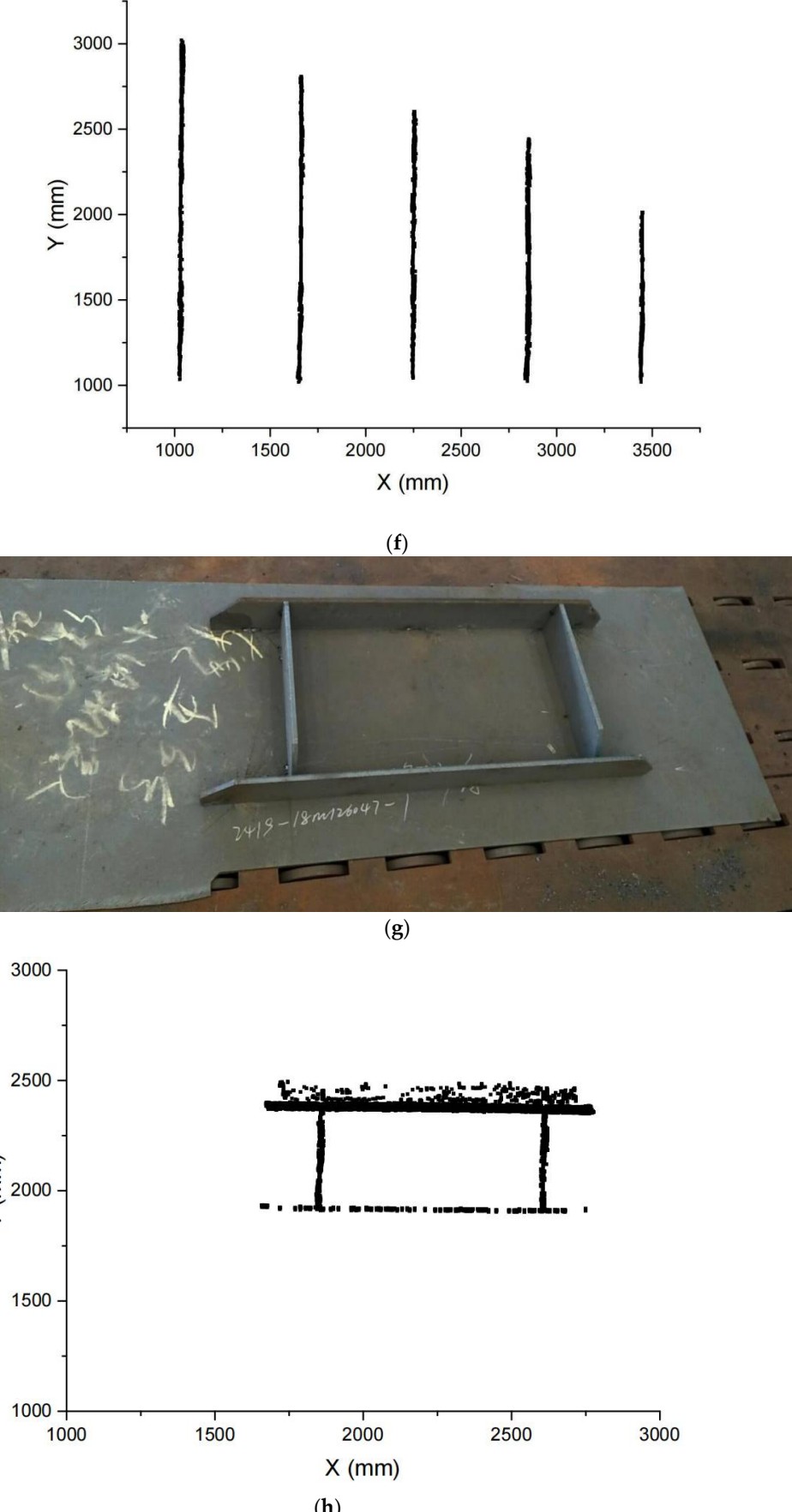

(f)

(g)

(h)

**Figure 6.** Small assemblies (**a,c,e,g**) and corresponding spatial points (**b,d,f,h**).

### 3.2. Grid-Based Line Segment Detector

Problems arise as we are trying to reconstruct all the stiffeners from these spatial points. Inevitable noise interferes with conventional line segment detectors (e.g., Hough transformation (HT), RANSAC, LSD). For example, line segments detected by the HT from spatial points of Figure 6b are marked in red in Figure 7. It's clear that noise results in missing points and false detection. Deliberately adjusting the parameters of the HT may alleviate the interference of noise. However, this is impractical in real-time manufacturing because the entire process is expected to run unsupervised. Moreover, the uncertainty of stiffeners' count, multi-density of points, and complex topology of small assemblies add more difficulties to this work.

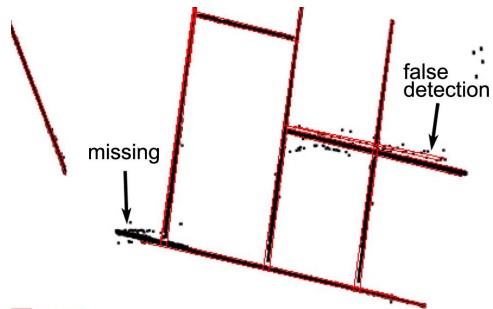

**Figure 7.** Line segments were detected (in red) from spatial points.

Thus, a new unsupervised approach is required to detect all of the line segments from spatial points. Here we introduce a grid-based line segment detector. It consists of 5 steps: sampling, rotation, convolution, deconvolution, and reconstruction, which are shown in Figure 8.

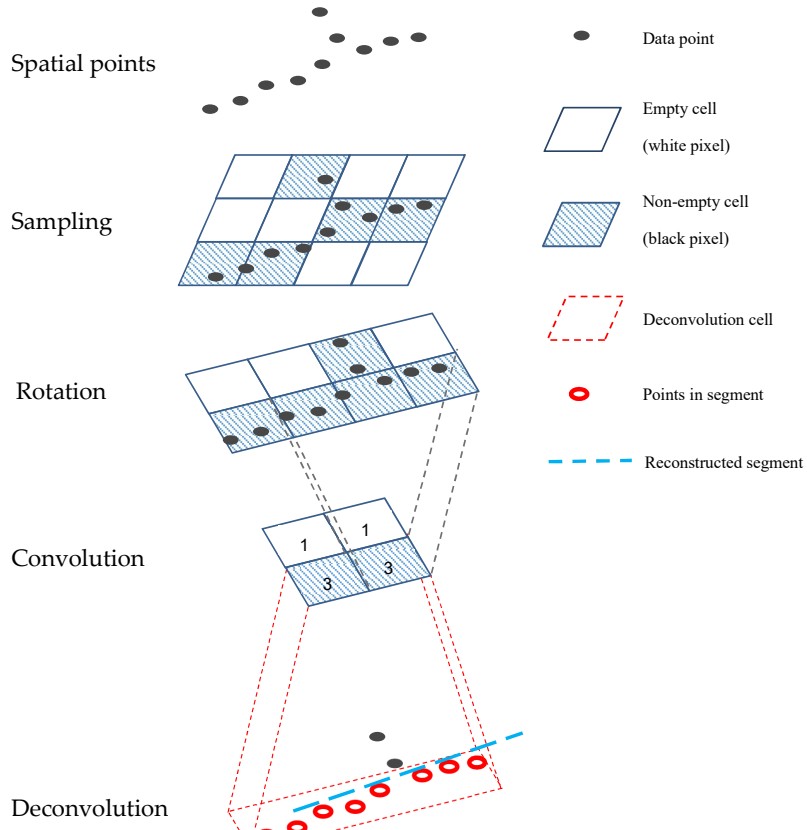

**Figure 8.** Steps of line segment reconstruction from spatial points.

Before we take the small assembly in Figure 6b to explain the working process of the detector, it is necessary to introduce the coordinate system on the workpiece. The X-axis was parallel to the gantry rails, the Y-axis was perpendicular to the rail. The system origin is located at the corner of the panel, as shown in Figure 9.

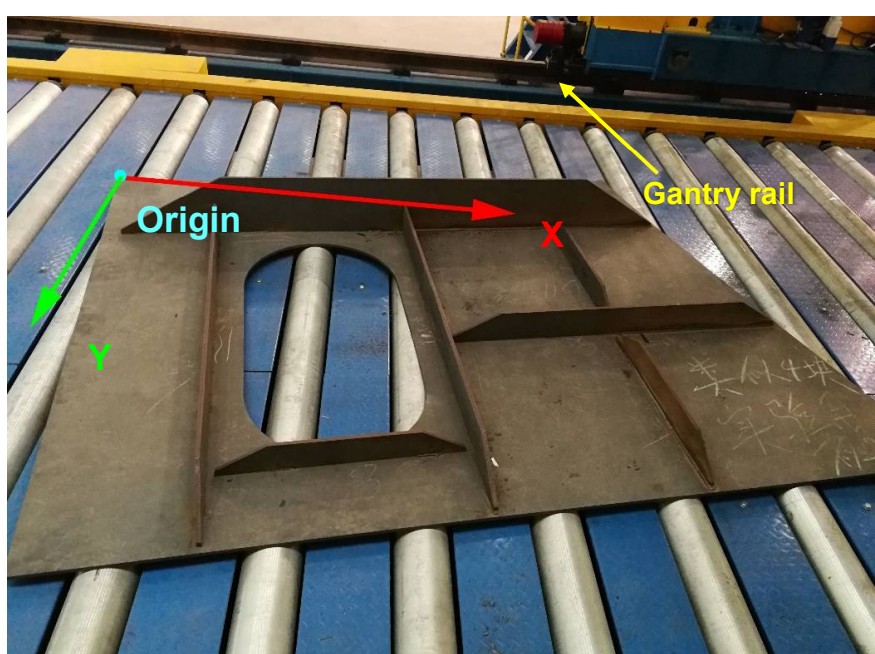

**Figure 9.** Coordinate system on the workpiece.

The first step is sampling, which greatly reduces the complexity of the following computation. All spatial points are projected into a grid consisting of square cells, and this is also the grid-based line segment detector named by. Marking empty cells as white, and non-empty cells as black, the grid could be converted to a binary feature map. Figure 10 shows the process of sampling 15,066 points from Figure 6b to a feature map of 78 × 63 cells.

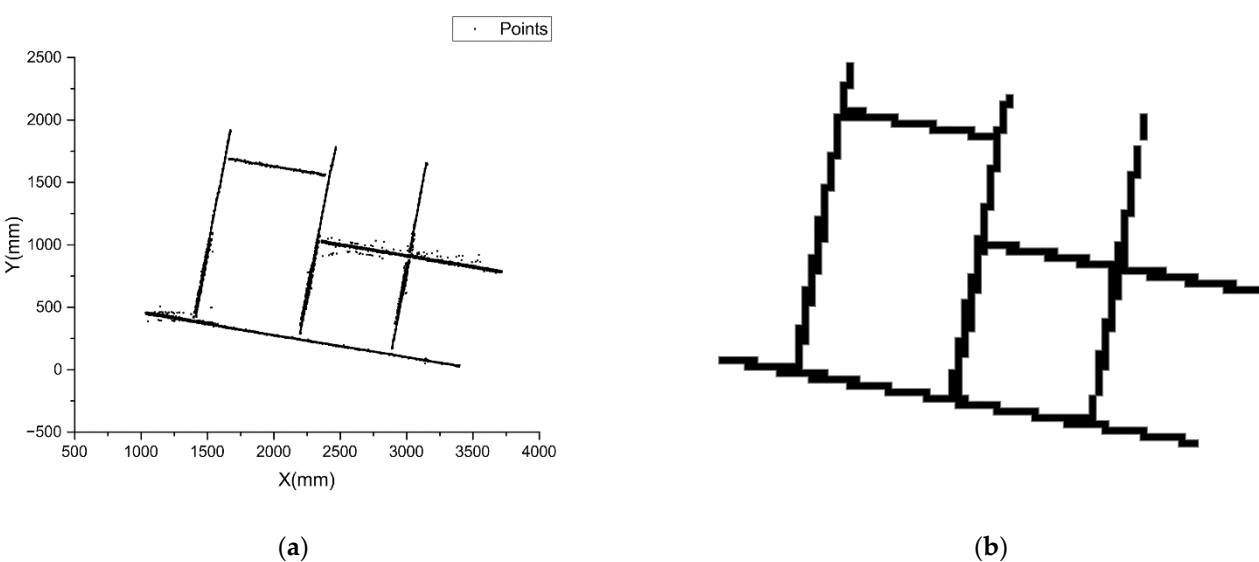

(**a**)                                             (**b**)

**Figure 10.** (**a**) Spatial Points (15,066 points); (**b**) Sampled feature map.

The rotation layer aligns the grid's axis to the point's major direction. The rotation angle of the axis is given by random sample consensus (RANSAC). RANSAC determines a single line from these points and calculates its angle from the x-axis, which is denoted by

$\theta$. The grid axis is then rotated by $\theta$ about the origin. Figure 11a shows the results of the rotation. After rotation, spatial points are re-sampled, as shown in Figure 11b.

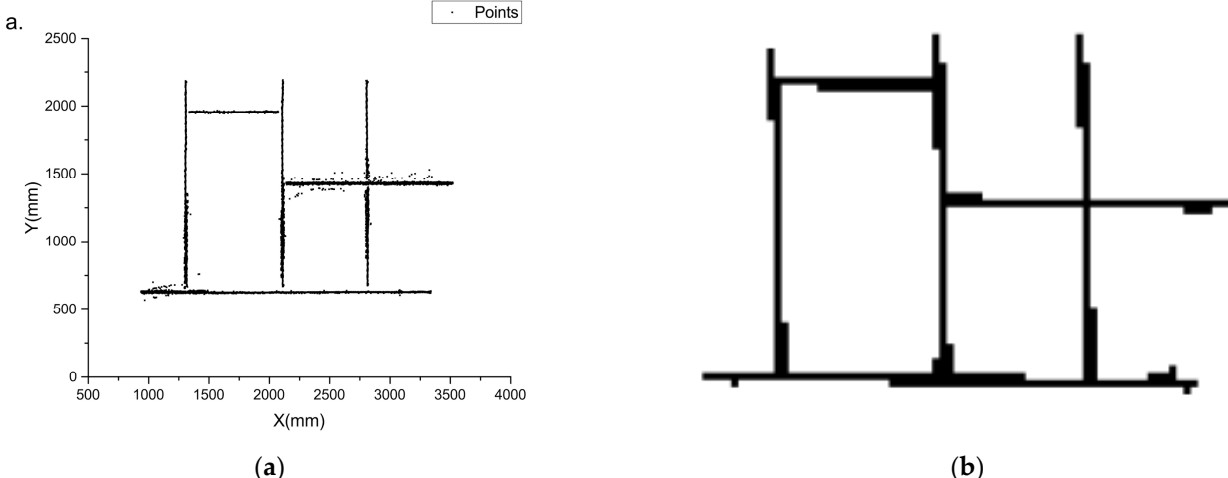

(a)

(b)

**Figure 11.** (a) Points after axis rotation; (b) Re-sampled feature map.

Re-sampled feature map is converted to a matrix $\{f(i, j)\}$, where the matrix entries $f(i, j) = 1$ for black pixels and $f(i, j) = 0$ for white pixels. Matrix $\{f(i, j)\}$ is convoluted to detect all line segments. Here 2 kernels are designed to detect horizontal and vertical segments.

First, the horizontal segment detector $K_h$ [size: $3 \times (2n + 1)$] is introduced, where $n$ equals half the length of the desired horizontal segment. The convolution operation using $K_h$ differs slightly from the conventional convolution operation. It consists of the following steps:

1.  Move kernel's center to a black pixel $(i,j)$, $i$ is the row index, $j$ is the column index
2.  $sum = 0$;
3.  $l = j - n$;
4.  Check pixels at $(i - 1,l)$, $(i,l)$ and $(i + 1,l)$, increase $sum$ by $1$ if any of these pixels is black;
5.  Increase $l$ by $1$;
6.  Repeat 3. until $l = j + n$;
7.  Convolution output at pixel $g(i,j) = 1$ if $sum$ is not less than $2n$, or $g(i,j) = 0$;
8.  Repeat 1. until all nonzero entries are convoluted.

Figure 12a shows the convolution output of Figure 10b, which removes all vertical segments. All horizontal segments could be recovered through the deconvolution process, as shown in Figure 12b. Finally, it helps us extract spatial points to reconstruct stiffeners in this direction, as shown in Figure 12c.

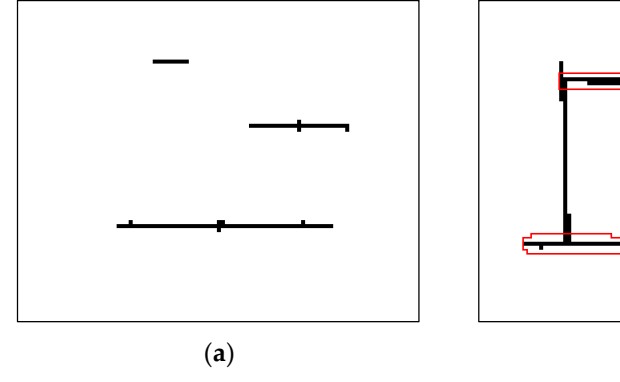
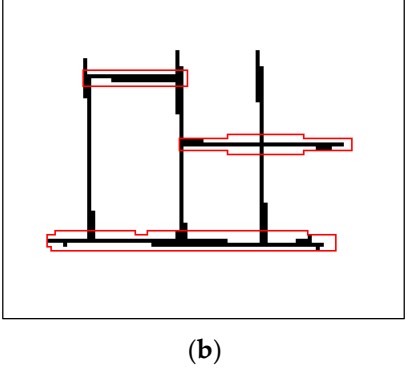
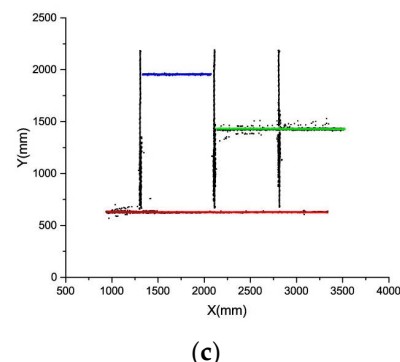

(a)

(b)

(c)

**Figure 12.** (a) Convolution output with $K_h$; (b) Horizontal segments recovered (in red bounds); (c) Points of each stiffener (in red, green, and blue).

Similarly, the vertical segments are detected by introducing kernel $K_v$. The size of $K_v$ is $(2m + 1) \times 3$, where $m$ is half the length of the desired vertical segment. The convolution using $K_v$ is similar to the algorithm used for the horizontal segments. Figure 13 shows the vertical segment detection process with kernel $K_v$.

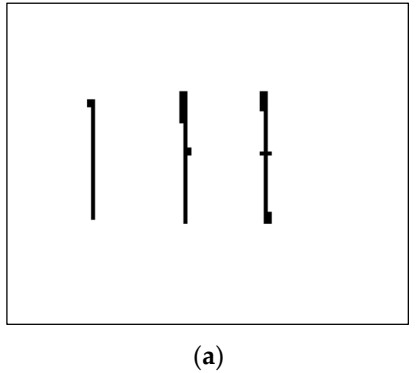

(a)

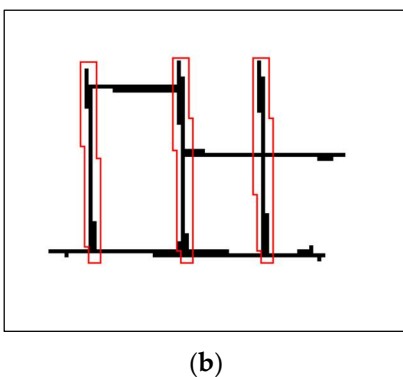

(b)

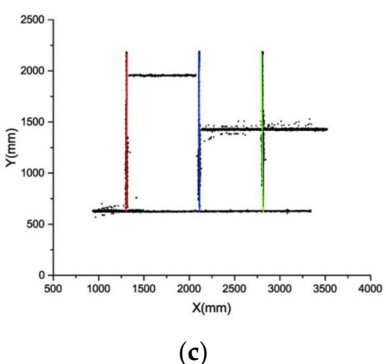

(c)

**Figure 13.** (**a**) Convolution output with $K_v$; (**b**) Vertical segments recovered (in red bounds); (**c**) Points of each stiffener (in red, green, and blue).

The last step is reconstructing all line segments. As the spatial points are classified into groups corresponding to each stiffener, the Least-Squares Fitting could perform this job without any challenge. Figure 14 shows the centerlines of all stiffeners by combing the results shown in Figures 12c and 13c. More ever, coordinates of stiffeners' free ends are listed in Table 1.

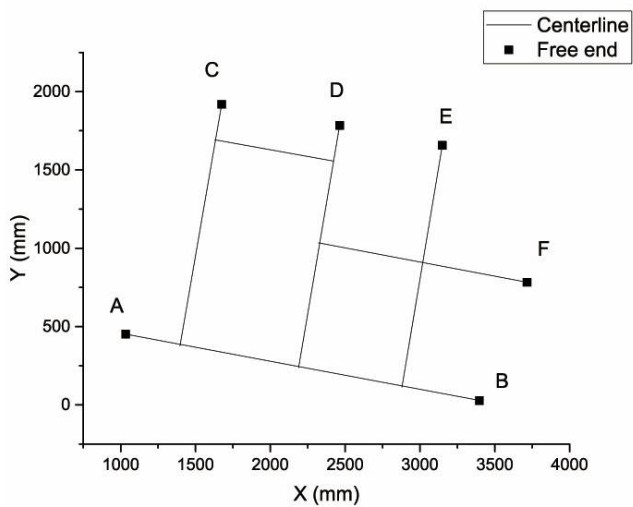

**Figure 14.** Reconstruction of all stiffeners on small assembly.

**Table 1.** Coordinates of the reconstructed free ends.

| Free End | X (mm) | Y (mm) |
|:---:|:---:|:---:|
| A | 1043 | 452 |
| B | 3431 | 28 |
| C | 1691 | 1919 |
| D | 2489 | 1784 |
| E | 3181 | 1656 |
| F | 3754 | 783 |

## 4. Results and Verification

To investigate the accuracy of the segmented detector, The coordinates of the stiffeners' free ends were manually measured and compared with the reconstructed position, as shown in Table 2.

**Table 2.** Distance between reconstructed and measured positions.

| Free End | Measured Position (x, y) | Reconstructed Position (x, y) | Distance (mm) |
|----------|--------------------------|-------------------------------|---------------|
| A | (1032, 426) | (1043, 452) | 28.2 |
| B | (3410, 25) | (3431, 28) | 21.2 |
| C | (1682, 1910) | (1691, 1919) | 12.7 |
| D | (2483, 1773) | (2489, 1784) | 10.8 |
| E | (3177, 1650) | (3181, 1656) | 7.2 |
| F | (3725, 798) | (3754, 783) | 32.6 |

In this study, the welding robot equips a range sensor for precise location and welding tracking. Tolerance of stiffener position is expected to be less than 50 mm. As shown in the last column of Table 2, all of the distances were less than 50 mm, satisfying the accuracy requirement.

Based on the demands of ship manufacturing, this algorithm is expected to give a reliable output, regardless of the kernel size or the shapes of the spatial points.

Figure 15a shows spatial points for which the convolutional grid-based clustering approach was adopted to extract all of the horizontal segments from the noise and other segments. Figure 15b shows the feature map sampled from these spatial points, and Figure 15c–e shows the convolution output for different sizes of $K_h$ ($n$ = 3, 4, 5). Figure 15c–e shows similar clustering numbers and positions of the black pixels, which indicated that the convolution output was insensitive to the value of $n$. As the goal of this step is to determine the number and positions of potential straight segments, this finding indicated the good robustness and tolerance of the algorithm, which are important for industrial applications. Thus, all the horizontal segments could be easily extracted using linear fitting. The results of the extracted points and segments are shown in Figure 15f.

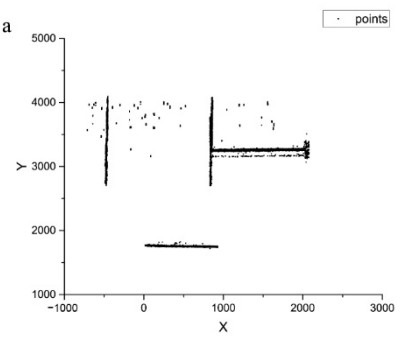
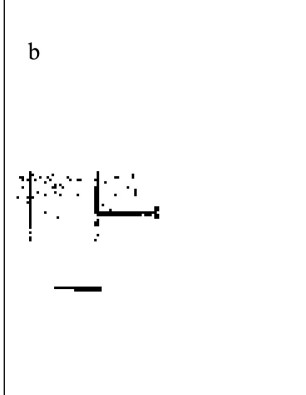

**Figure 15.** *Cont.*

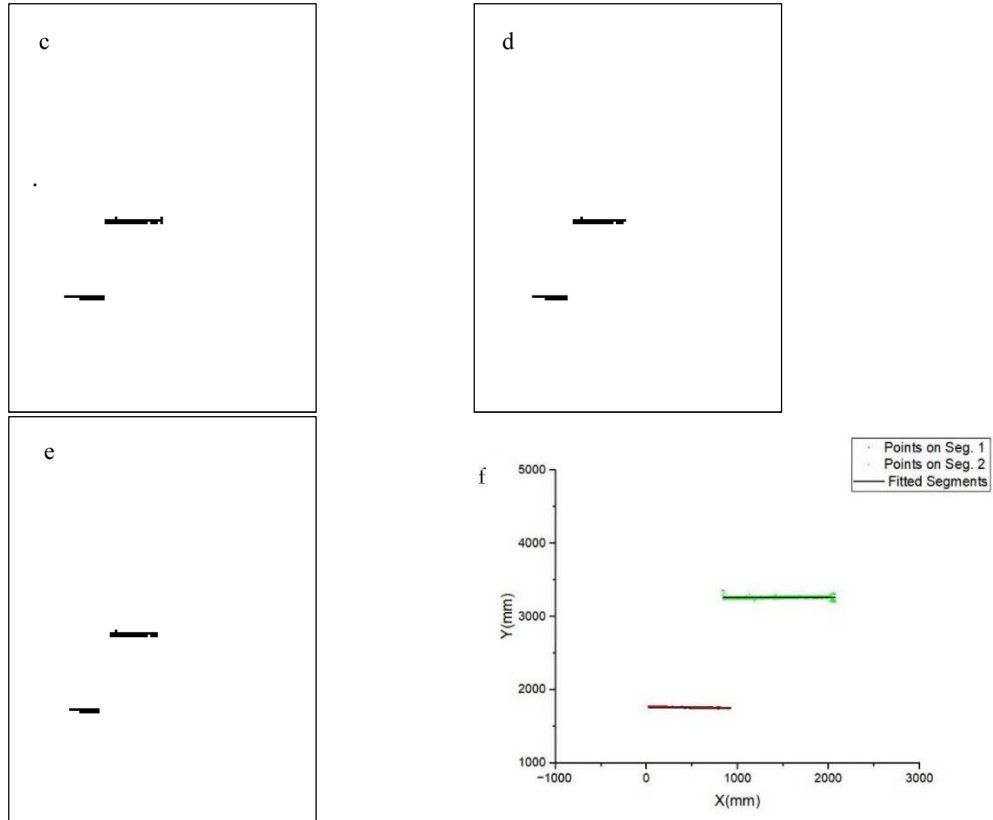

**Figure 15.** (**a**) Spatial points of small assemblies; (**b**) Feature map; (**c**–**e**). Convolution output with $K_h$ (*n* = 3, 4, and 5, respectively); (**f**) Stiffener center line reconstructed.

The line segment detector is expected to deal with small assemblies of various shapes in the shipyard, thus spatial points of small assemblies from actual manufacturing are tested. Figure 16a,c,e show the spatial points, and Figure 16b,d,f show the corresponding segments reconstruction results. The comparison of the spatial points and reconstructed segments shows perfect matches.

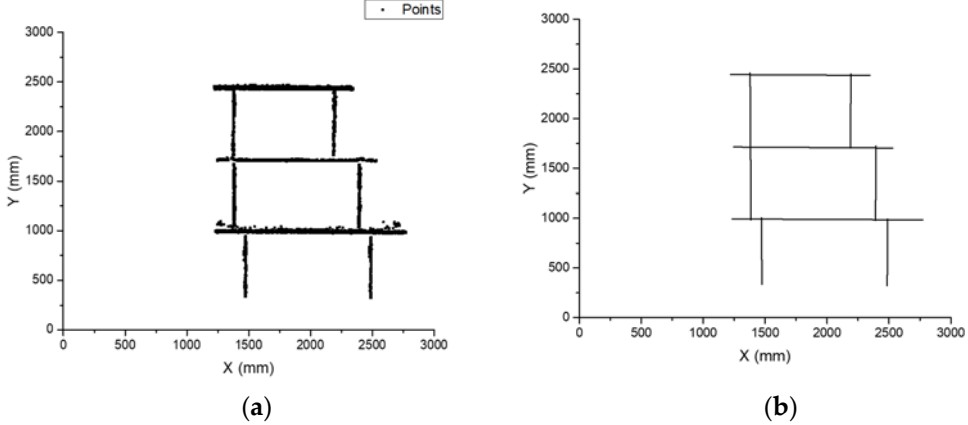

**Figure 16.** *Cont.*

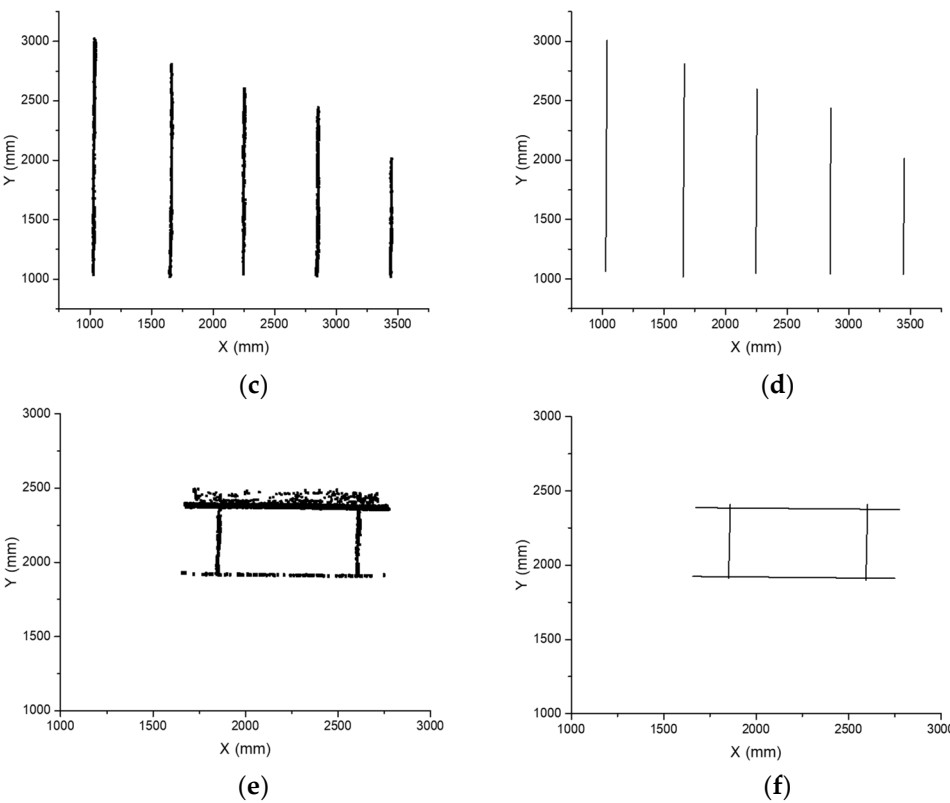

**Figure 16.** (**a**,**c**,**e**) Spatial points of small assemblies; (**b**,**d**,**f**) Reconstructed segments of stiffeners.

## 5. Application

Figure 17 describes a general working flow of autonomous ship small assembly line, in which the line segment detector plays a crucial role. Data volume from the TOF sensor ranges from 500 MB to 5 GB, and spatial points number are from 50,000 to 100,000. The grid-based algorithm contributes to reducing computation complexity, allowing the assembly line welding to multiply workpieces simultaneously.

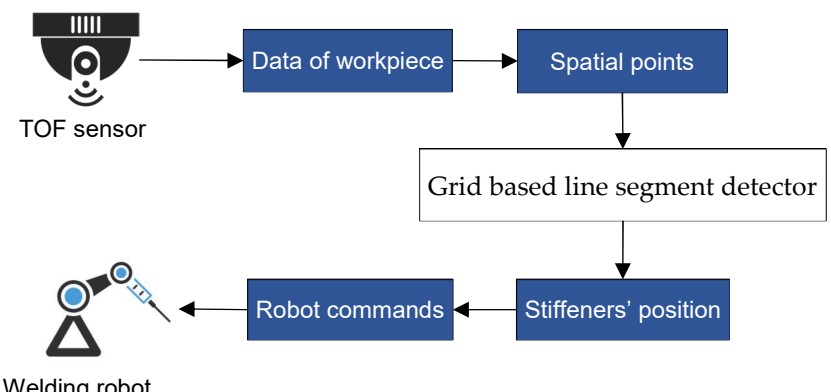

**Figure 17.** Working flow of autonomous ship small assembly line.

The line segment detector was implemented on an autonomous ship small assembly line in Guangzhou Shipyard International Company Ltd., belonging to China State Shipbuilding Corporation Ltd. (CSSC), as shown in Figure 18. The assembly line has been put into production since 2019. The vision system guided robots preforming various welding jobs without manual teaching or programming. It is also shown that the assembly line is capable of mass production, taking the leading position in this field.

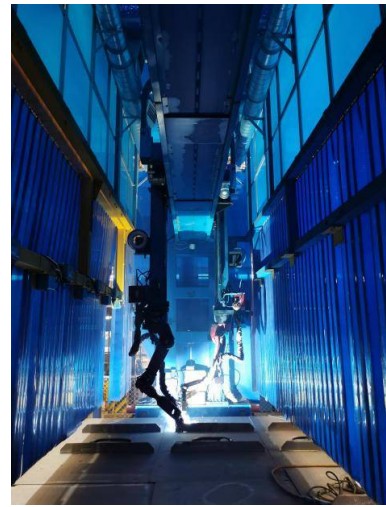

(**a**)

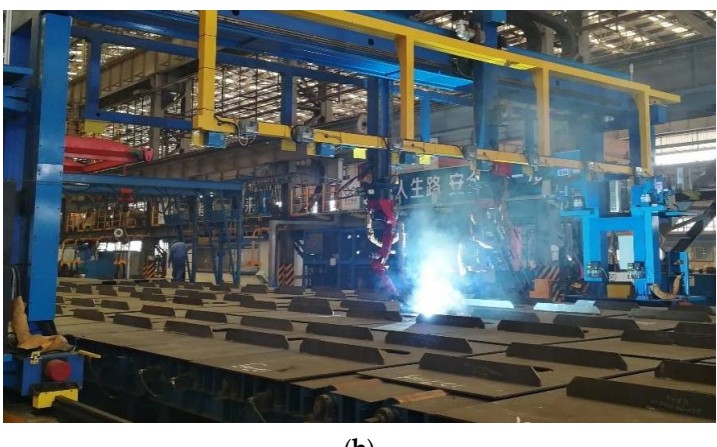

(**b**)

**Figure 18.** (**a**) Vision guided welding robots in production at shipyard; (**b**) Overview of the assembly line.

## 6. Conclusions

In this paper, a grid-based line segment detector was introduced, which was used to guide robots for autonomous welding in ship-building. Intelligent robotic welding systems were implemented using the proposed algorithm, and the approach was verified under actual manufacturing conditions. The contributions of this study are summarized as follows:

(1) The method presented in this paper demonstrated good robustness. It successfully clustered the points of stiffeners with complicated structures, despite the interference of noise. The robustness is a great advantage for industrial applications.

(2) This algorithm was verified under manufacturing conditions and exhibited accuracy and robustness. Based on this work, intelligent ship small assembly lines were implemented and put into production in shipyards.

Curved segments, which are also important in ship-building, were not considered in this study. In future research, a detector for curved segments will be conducted to fill the gaps of this study. Also, we are planning to extend our research to more cases in shipbuilding, like welding jobs for ship blocks, reversing modeling of hull structures, etc.

**Author Contributions:** Conceptualization, C.N.; software, C.N.; validation, J.D.; writing—original draft preparation, J.D.; writing—review and editing, C.N.; visualization, J.D.; project administration, C.N. All authors have read and agreed to the published version of the manuscript.

**Funding:** This research received no external funding.

**Institutional Review Board Statement:** Not applicable.

**Informed Consent Statement:** Not applicable.

**Data Availability Statement:** Not applicable.

**Conflicts of Interest:** The authors declare no conflict of interest.

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
