# Peer review of "Gird Based Line Segment Detector and Application: Vision System for Autonomous Ship Small Assembly Line"

_jmse, doi:10.3390/jmse9111313_

Round 1
Reviewer 1 Report
Gird Based Line Segment Detector and Application: Vision System for Autonomous Ship Small Assembly Line
Several parts need to be clarified:
- The algorithm/script has not been explained. It is mentioned in Table 1 that there are comparisons between measured and reconstructed points. How is the algorithm composed?
- Related no. 1, how was the measured point obtained? Clarify the procedure to measure the points.
- The application is not clear. Application has to clarify the performance of the composed algorithm in executing the real-world tasks, e.g. welding.
- A literature review needs to be composed. What is the main motive or urgency so that this technology is needed?
- How is the milestone in welding so that collaboration with this technology is needed?
- Authors need to clarify methodology and results more explicitly. When I read this paper is like the authors presented a technology without further scientific explanation regarding script/algorithm data/or any details regarding the development of this technology.
Author Response
Point 1:The algorithm/script has not been explained. It is mentioned in Table 1 that there are comparisons between measured and reconstructed points. How is the algorithm composed?
Response 1: Complement of algorithm is added in section 3.2. Now the algorithm composes of 5 steps: sampling, rotation, convolution, de-convolution and reconstruction. We give a more detailed explaination in this part.
Point 2: Related no. 1, how was the measured point obtained? Clarify the procedure to measure the points.
Response 2: These points are manually measured. We add the introduce of coordinate system on workpiece in section 3.2.
Point 3: The application is not clear. Application has to clarify the performance of the composed algorithm in executing the real-world tasks, e.g. welding.
Response 3: How the algorithm works in real-world tasks is added in section 5.
Point 4: A literature review needs to be composed. What is the main motive or urgency so that this technology is needed?
Response 4: Literature review is enriched with some latest archives. The main motive is to solve a common difficulties in intelligent ship manufacture.
Point 5: How is the milestone in welding so that collaboration with this technology is needed?
Response 5: Algorithm in this study shows great advantage in data compressing, making it possible for mass production. Assembly line developed on this papers' work is capable of welding multiply workpieces simultaneously, leading similar products from competitors (e.g. Inro. Tech. robots)
Point 6: Authors need to clarify methodology and results more explicitly. When I read this paper is like the authors presented a technology without further scientific explanation regarding script/algorithm data/or any details regarding the development of this technology.
Response 6: We supply more technical details in revised manuscript, wishing it meet the expectation.
Reviewer 2 Report
Dear Authors,
After improving the paper including the comments previously suggested, there are not further changes required.
Kind regards,
Author Response
Point 1: After improving the paper including the comments previously suggested, there are not further changes required.
Response: The manuscript is improved, thanks for the suggestion.
Reviewer 3 Report
The article introduces the principle and technique details of a vision system which guides welding robot in ship small assembly production. A new unsupervised line segment detector is proposed to reconstruct ship small assemblies from spatial points.
Some clarifications are needed and some recommendations for improvements are given below.
Line 66, it is recommended to give some dimensional and mass description on what is meant under small assemblies
Line 70, the features of the robot should be stated. Also, the figure is too small and it shoud be in better resolution. Some interesting parts of figure should be additionaly magnified as a separate figure.
Line 75, figure too small
Line 93, Figure 5, should be given more explanation
Line 98, Figure 6, needs more explanation
Line 102, Figure 6 should be better explained in the text
Line 114, Figure 7, The chosen colors are not giving the immediate expression due to explanation given in the text.
Line 122, Figure 9, The coordinate system is missing on 9 (b), it is difficult to say how good it is, suggested method, without it
Line129, Figure 10, same as Figure 10 (b)
Line 170, the value of 50 mm seems to be to high in the welding process!
Line 188, the coordinate system is missing
Line 206, too small figure, how can we know that the robots are guided by vision system and that they are working without manual teaching or programming?
Generally, all figures should be better explained in the text.
As it is promised at the beginning of the article the good job is done but not explained very well.
The good tolerance, robustness and accuracy of the method is not well demonstrated.
Author Response
Point 1: Line 66, it is recommended to give some dimensional and mass description on what is meant under small assemblies
Response 1: General description of small assemblies are add.
Point 2: Line 70, the features of the robot should be stated. Also, the figure is too small and it shoud be in better resolution. Some interesting parts of figure should be additionaly magnified as a separate figure.
Response 2: We replaced the figure with its raw picture. We also add a picture of the TOF sensor which is very important for this study.
Point 3: Line 75, figure too small
Response 3: The figure is replaced.
Point 4: Line 93, Figure 5, should be given more explanation
Response 4: Explanation of laser stripe and scanned points is added, in text and figure.
Point 5: Line 98, Figure 6, needs more explanation
Response 5: These figures are replaced by raw pictures, and explanation is added, seeing response 4.
Point 6: Line 102, Figure 6 should be better explained in the text
Response 6: Relationship of workpiece and spatial points are now explained in Section 3.1, seeing response 4.
Point 7: Line 114, Figure 7, The chosen colors are not giving the immediate expression due to explanation given in the text.
Response 7: We added the explanation of figure 7, segments in red are the output from standard hough-transform.
Point 8: Line 122, Figure 9, The coordinate system is missing on 9 (b), it is difficult to say how good it is, suggested method, without it
Response 8: We add the explanation of coordinate system on workpiece. For 9(b), it is a magnified binary- image, thus it employs a pixel coordinate system, which is quite different from 9(a). I believe it is unnecessary to compare the two coordinates system.
Point 9: Line129, Figure 10, same as Figure 10 (b)
Response 9: Seeing response 8, the pixel coordinate system in 10(b) is quite different from it in 10(a).
Point 10: Line 170, the value of 50 mm seems to be to high in the welding process!
Response 10: The explanation is supplied at the start of Section 4. The welding robot equips with range sensor for precise location and welding tracking. Tolerance of stiffener position is expected to be less than 50 mm.
Point 11: Line 188, the coordinate system is missing
Response 11: Seeing response 8. The spatial points and feature map employs different coordinate system. Comparison between 2 system is not encouraged.
Point 12: Line 206, too small figure, how can we know that the robots are guided by vision system and that they are working without manual teaching or programming?
Response 12: Now greater picture of manufacturing is provided. It is of quite difficulty to prove that the robots are guided by vision by static pictures. The assembly line has been put into production for 3 years, videos, production records and many other documents could be evidence but not suitable for publication.
Point 13: Generally, all figures should be better explained in the text. As it is promised at the beginning of the article the good job is done but not explained very well. The good tolerance, robustness and accuracy of the method is not well demonstrated.
Response 13: The manuscript is supplemented with better figures, more technical details, hopeful for the expectation.
Round 2
Reviewer 1 Report
The comments have been well addressed.
Reviewer 3 Report
-